# NeMo: Neural Mesh Models of Contrastive Features for Robust 3D Pose Estimation

**Angtian Wang, Adam Kortylewski, Alan Yuille**
Department of Computer Science
Johns Hopkins University
Maryland, MD 21218, USA
{angtianwang, akortyl1, ayuille1}@jhu.edu

## Abstract

3D pose estimation is a challenging but important task in computer vision. In this work, we show that standard deep learning approaches to 3D pose estimation are not robust when objects are partially occluded or viewed from a previously unseen pose. Inspired by the robustness of generative vision models to partial occlusion, we propose to integrate deep neural networks with 3D generative representations of objects into a unified neural architecture that we term NeMo. In particular, NeMo learns a generative model of neural feature activations at each vertex on a dense 3D mesh. Using differentiable rendering we estimate the 3D object pose by minimizing the reconstruction error between NeMo and the feature representation of the target image. To avoid local optima in the reconstruction loss, we train the feature extractor to maximize the distance between the individual feature representations on the mesh using contrastive learning. Our extensive experiments on PASCAL3D+, occluded-PASCAL3D+ and ObjectNet3D show that NeMo is much more robust to partial occlusion and unseen pose compared to standard deep networks, while retaining competitive performance on regular data. Interestingly, our experiments also show that NeMo performs reasonably well even when the mesh representation only crudely approximates the true object geometry with a cuboid, hence revealing that the detailed 3D geometry is not needed for accurate 3D pose estimation. The code is publicly available at https://github.com/Angtian/NeMo.

## 1 Introduction

Object pose estimation is a fundamentally important task in computer vision with a multitude of real-world applications, e.g. in self-driving cars, or partially autonomous surgical systems. Advances in the architecture design of deep convolutional neural networks (DCNNs) Tulsiani & Malik (2015); Su et al. (2015); Mousavian et al. (2017); Zhou et al. (2018) increased the performance of computer vision systems at 3D pose estimation enormously. However, our experiment shows current 3D pose estimation approaches are not robust to partial occlusion and when objects are viewed from a previously unseen pose. This lack of robustness can have serious consequences in real-world applications and therefore needs to be addressed by the research community.

In general, recent works follow either of two approaches for object pose estimation: Keypoint-based approaches detect a sparse set of keypoints and subsequently align a 3D object representation to the detection result. However, due to the sparsity of the keypoints, these approaches are highly vulnerable when the keypoint detection result is affected by adverse viewing conditions, such as partial occlusion. On the other hand, rendering-based approaches utilize a generative model, that is built on a dense 3D mesh representation of an object. They estimate the object pose by reconstructing the input image in a render-and-compare manner (Figure 1). While rendering-based approaches can be more robust to partial occlusion Egger et al. (2018), their core limitation is that they model objects in terms of image intensities. Therefore, they pay too much attention to object details that are not relevant for the 3D pose estimation task. This makes them difficult to optimize Blanz & Vetter (2003); Schönborn et al. (2017), and also requires a detailed mesh representation for every shape variant of an object class (e.g. they need several types of sedan meshes instead of using one prototypical type of sedan).

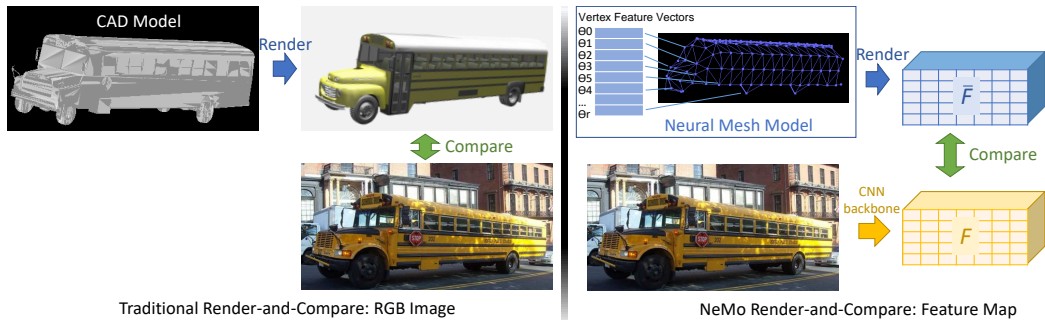

Figure 1: Traditional render-and-compare approaches render RGB images and make pixel-level comparisons. These are difficult to optimize due to the many local optima in the pixel-wise reconstruction loss. In contrast, NeMo is a Neural Mesh Model that renders feature maps and compares them with feature maps obtained via CNN backbone. The invariance of the neural features to nuisance variables, such as shape and color variations, enables a robust 3D pose estimation with simple gradient-descent optimization of the neural reconstruction loss.

In this work, we introduce NeMo a rendering-based approach to 3D pose estimation that is highly robust to partial occlusion, while also being able to generalize to previously unseen views. Our key idea is to learn a generative model of an object category in terms of neural feature activations, instead of image intensities (Figure 1). In particular, NeMo is composed of a prototypical mesh representation of the object category and feature representations at each vertex of the mesh. The feature representations are learned to be invariant to instance specific details (such as shape and color variations) that are not relevant for the 3D pose estimation task. Specifically, we use contrastive learning He et al. (2020); Wu et al. (2018); Bai et al. (2020) to ensure that the extracted features of an object are distinct from each other (e.g. the features of the front tire of a car are different from those of the back tire), while also being distinct from non-object features in the background. Furthermore, we train a generative model of the feature activations at every vertex of the mesh representation. During inference, NeMo estimates the object pose by reconstructing a target feature map with using render-and-compare and gradient-based optimization w.r.t. the 3D object pose parameters.

We evaluate NeMo at 3D pose estimation on the PASCAL3D+ Xiang et al. (2014) and the ObjectNet3D Xiang et al. (2016) dataset. Both datasets contain a variety of rigid objects and their corresponding 3D CAD models. Our experimental results show that NeMo outperforms popular approaches such as Starmap Zhou et al. (2018) at 3D pose estimation by a wide margin under partial occlusion, and performs comparably when the objects are not occluded. Moreover, NeMo is exceptionally robust when objects are seen from a viewpoint that is not present in the training data. Interestingly, we also find that the mesh representation in NeMo can simply approximate the true object geometry with a cuboid, and still perform very well. Our main contributions are:

1. We propose a 3D neural mesh model of objects that is generative in terms of contrastive neural network features. This representation combines a prototypical geometric representation of the object category with a generative model of neural network features that are invariant to irrelevant object details.

2. We demonstrate that standard deep learning approaches to 3D pose estimation are highly sensitive to out-of-distribution data including partial occlusions and unseen poses. In contrast, NeMo performs 3D pose estimation with exceptional robustness.

3. In contrast to other rendering-based approaches that require instance-specific mesh representations of the target objects, we show that NeMo achieves a highly competitive 3D pose estimation performance even with a very crude prototypical approximation of the object geometry using a cuboid.

## 2   RELATED WORK

**Category-Level Object Pose Estimation.**   Category-Level object pose estimation has been well explored by the research community. A classical approach as proposes by Tulsiani & Malik (2015) and Mousavian et al. (2017) was to formulate object pose estimation as a classification problem. Another common category-level object pose estimation approach involves a two-step process Szeto & Corso (2017); Pavlakos et al. (2017): First, semantic keypoints are detected interdependently and subsequently a Perspective-n-Point problem is solved to find the optimal 3D pose of an object mesh

Lu et al. (2000); Lepetit et al. (2009). Zhou et al. (2018) further improved this approach by utlizing depth information. Recent work Wang et al. (2019); Chen et al. (2020) introduced render-and-compare to for category-level pose estimation. However, both approaches used pixel-level image synthesis and required detailed mesh models during training.In contrast, NeMo preforms render-and-compare on the level of contrastive features, which are invariant to intra-category nuisances, such as shape and color variations. This enables NeMo to achieve accurate 3D pose estimation results even with a crude prototypical category-level mesh representation.

**Pose Estimation under Partial Occlusion.** Keypoint-based pose estimation methods are sensitive to outliers, which can be caused by partial occlusion Pavlakos et al. (2017); Sundermeyer et al. (2018). Some rendering-based approaches achieve satisfactory results on instance-level pose estimation under partial occlusion Song et al. (2020); Peng et al. (2019); Zakharov et al. (2019); Li et al. (2018). However, these approaches render RGB images or use instance-level constraints, e.g. pixel-level voting, to estimate object pose. Therefore, these approaches are not suited for category-level pose estimation. To the best of our knowledge, NeMo is the first approach that performs category-level pose estimation robustly under partial occlusion.

**Contrastive Feature Learning.** Contrastive learning is widely used in deep learning research. Hadsell et al. (2006) proposed an intuitive tuple loss, which was later extended to triplets and N-Pair tuples Schroff et al. (2015); Sohn (2016). Recent contrastive learning approaches showed high potential in unsupervised learning Wu et al. (2018); He et al. (2020). Oord et al. (2018); Han et al. (2019) demonstrated the effectiveness of feature-level contrastive losses in representation learning. Bai et al. (2020) proposed a keypoint detection framework by optimizing the feature representations of keypoints with a contrastive loss. In this work, we use contrastive feature learning to encourage the feature extraction backbone to extract locally distinct features. This, in turn, enables 3D pose estimation by simply optimizing the neural reconstruction loss with gradient descent.

**Robust Vision through Analysis-by-Synthesis.** In a broader context, our work relates to a line of work in the computer vision literature, which demonstrate that the explicit modeling of the object structure significantly enhances the robustness of computer vision models, e.g. at 3D pose estimation Zeeshan Zia et al. (2013), face reconstruction Egger et al. (2018) and human detection Girshick et al. (2011) under occlusion. More specifically, our work builds on a recent line of work that introduced a neural analysis-by-synthesis approach to vision Kortylewski et al. (2020b) and demonstrated its effectiveness in occlusion-robust image classification Kortylewski et al. (2020a;c) and object detection Wang et al. (2020). Our work significantly extends neural analysis-by-synthesis to include an explicit 3D object representation, instead of 2D template-like object representations. This enables our model to achieve state-of-the-art robustness at pose estimation through neural render-and-compare.

## 3 NeMo: A 3D generative model of neural features

We denote a feature representation of an input image $I$ as $\Phi(I) = F^l \in \mathbb{R}^{H \times W \times D}$. Where $l$ is the output of a layer $l$ of a DCNN $\Phi$, with $D$ being the number of channels in layer $l$. $f_i^l \in \mathbb{R}^D$ is a feaure vector in $F^l$ at position $i$ on the 2D lattice $\mathcal{P}$ of the feature map. In the remainder of this section we omit the superscript $l$ for notational simplicity because this is fixed a-priori in our model.

### 3.1 Neural Rendering of Feature Maps

Similar to other graphics-based generative models, such as e.g. 3D morphable models Blanz & Vetter (1999); Egger et al. (2018), our model builds on a 3D mesh representation that is composed of a set of 3D vertices $\Gamma = \{r \in \mathbb{R}^3 | r = 1, \ldots, R\}$. For now, we assume the object mesh to be given at training time but we will relax this assumption in later sections. Different from standard graphics-based generative models, we do not store RGB values at each mesh vertex $r$ but instead store feature vectors $\Theta = \{\theta_r \in \mathbb{R}^D | r = 1, \ldots, R\}$. Using standard rendering techniques, we can use this 3D neural mesh model $\mathfrak{N} = \{\Gamma, \Theta\}$ to render feature maps:

$$\bar{F}(m) = \Re(\mathfrak{N}, m) \in \mathbb{R}^{H \times W \times D}, \tag{1}$$

where $m$ are the camera parameters for projecting the neural mesh representation (Figure 2).

### 3.2 Neural Mesh Models

Neural Mesh Models are probabilistic generative models of neural feature activations. Hence, our goal is to learn a generative model $p(F|\mathfrak{N}_y)$ of the real-valued feature activations $F$ of an object

class $y$ by leveraging a 3D neural mesh representation $\mathfrak{N}_y$. Assuming that the 3D pose $m$ of the object in the input image is known, we define the likelihood of the feature representation $F$ as:

$$p(F|\mathfrak{N}_y, m, B) = \prod_{i \in \mathcal{FG}} p(f_i|\mathfrak{N}_y, m) \prod_{i' \in \mathcal{BG}} p(f_{i'}|B). \tag{2}$$

The foreground $\mathcal{FG}$ is the set of all positions on the 2D lattice $\mathcal{P}$ of the feature map $F$ that are covered by the rendered neural mesh model. We compute $\mathcal{FG}$ by projecting the 3D vertices of the mesh model $\Gamma_y$ into the image using the ground truth camera pose $m$ to obtain the 2D locations of the visible vertices in the image $\mathcal{FG} = \{s_t \in \mathbb{R}^2 | t = 1, \ldots, T\}$. We define foreground feature likelihoods to be Gaussian distributed:

$$p(f_i|\mathfrak{N}_y, m) = \frac{1}{\sigma_r \sqrt{2\pi}} \exp\left(-\frac{1}{2\sigma_r^2} \|f_i - \theta_r\|^2\right). \tag{3}$$

Note that the correspondence between the feature vector $f_i$ in the feature map $F$ and the vector $\theta_r$ on the neural mesh model is given by the 2D projection of $\mathfrak{N}_y$ with camera parameters $m$. Those features that are not covered by the neural mesh model $\mathcal{BG} = \mathcal{P} \setminus \{\mathcal{FG}\}$, i.e. are located in the background, are modeled by a Gaussian likelihood:

$$p(f_{i'}|B) = \frac{1}{\sigma \sqrt{2\pi}} \exp\left(-\frac{1}{2\sigma^2} \|f_{i'} - \beta\|^2\right), \tag{4}$$

with mixture parameters $B = \{\beta, \sigma\}$.

## 3.3 Training using Maximum Likelihood and Contrastive Learning

During training we want to optimize two objectives: 1) The parameters of the generative model as defined in Equation 2 should be optimized to achieve maxmimum likelihood on the training data. 2) The backbone used for feature extraction $\psi$ should be optimized to make the individual feature vectors as disctinct from each other as possible.

**Maximum likelihood estimation of the generative model.** We optimize the parameters of the generative model to minimize the negative log-likelihood of our model (Equation 2):

$$\mathcal{L}_{ML}(F, \mathfrak{N}_y, m, B) = -\ln p(F|\mathfrak{N}_y, m, B) \tag{5}$$

$$= -\sum_{i \in \mathcal{FG}} \ln\left(\frac{1}{\sigma_r \sqrt{2\pi}}\right) - \frac{1}{2\sigma_r^2} \|f_i - \theta_r\|^2 \tag{6}$$

$$+ \sum_{i' \in \mathcal{BG}} \ln\left(\frac{1}{\sigma \sqrt{2\pi}}\right) - \frac{1}{2\sigma^2} \|f_{i'} - \beta\|^2 \tag{7}$$

If we constrain the variances such that $\{\sigma^2 = \sigma_r^2 = 1 | \forall r\}$ then the maximum likelihood loss reduces to:

$$\mathcal{L}_{ML}(F, \mathfrak{N}_y, m, B) = -C \sum_{i \in \mathcal{FG}} \|f_i - \theta_r\|^2 + \sum_{i' \in \mathcal{BG}} \|f_{i'} - \beta\|^2, \tag{8}$$

where $C$ is a constant scalar.

**Contrastive learning of the feature extractor.** The general idea of the contrastive loss is to train the feature extractor such that the individual feature vectors on the object are distinct from each other, as well as from the background:

$$\mathcal{L}_{Feature}(F, \mathcal{FG}) = -\sum_{i \in \mathcal{FG}} \sum_{i' \in \mathcal{FG} \setminus \{i\}} \|f_i - f_{i'}\|^2 \tag{9}$$

$$\mathcal{L}_{Back}(F, \mathcal{FG}, \mathcal{BG}) = -\sum_{i \in \mathcal{FG}} \sum_{j \in \mathcal{BG}} \|f_i - f_j\|^2. \tag{10}$$

The contrastive feature loss $\mathcal{L}_{Feature}$ encourages the features on the object to be distinct from each other (e.g. the feature vectors at the front tire of a car are different from those of the back tire). The contrastive background loss $\mathcal{L}_{Back}$ encourages the features on the object to be distinct from the features in the background. The overall loss used to train NeMo is:

$$\mathcal{L}(F, \mathfrak{N}_y, m, B) = \mathcal{L}_{ML}(F, \mathfrak{N}_y, m, B) + \mathcal{L}_{Feature}(F, \mathcal{FG}) + \mathcal{L}_{Back}(F, \mathcal{FG}, \mathcal{BG}) \tag{11}$$

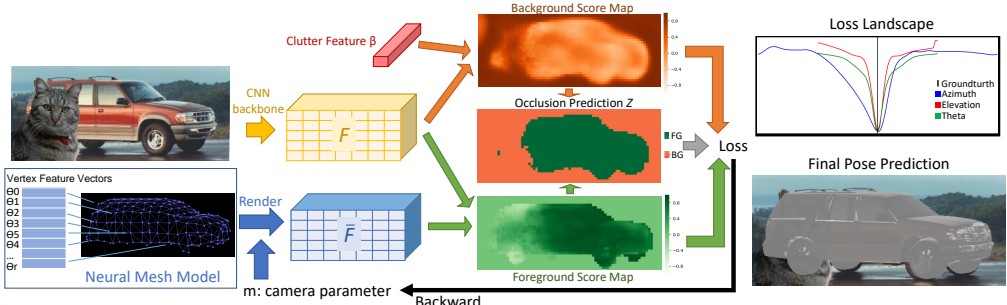

Figure 2: Overview of pose estimation: For each image, we use the trained CNN backbone to extract feature map $F$. Meanwhile, using trained Neural Mesh Model and randomly initialized object pose, we can render a feature map $\bar{F}$. By calculating similarity at each local of $F$ and $\bar{F}$, we can create a foreground score map, which demonstrate the object likelihood at each location. Similarly, we can get a background score map via $F$ and trained clutter model $\beta$. Using these two maps, we do the occlusion inference to segment image into foreground region and background region. Then, we calculate reconstruction loss and optimize object pose via minimize the loss. We also visualize the loss landscape along all 3 object pose parameters, and the final pose prediction.

## 3.4 Robust 3D Pose Estimation with Render and Compare

After training the feature extractor and the generative model in NeMo, we apply the model for estimating the camera pose parameters $b$. In particular we aim to optimize the model likelihood from Equation 2 w.r.t. to the camera parameters in a render-and-compare manner. Following related work on robust inference with generative models Kortylewski (2017); Egger et al. (2018) we optimize a robust model likelihood:

$$p(F|\mathfrak{N}_y, m, B, z_i) = \prod_{i \in \mathcal{FG}} \left[ p(f_i|\mathfrak{N}_y, m) p(z_i=1) \right]^{z_i} \left[ p(f_i|B) p(z_i=0) \right]^{(1-z_i)} \prod_{i' \in \mathcal{BG}} p(f_{i'}|B). \quad (12)$$

Here $z_i \in \{0, 1\}$ is a binary variable and $p(z_i=1)$ and $p(z_i=0)$ are the prior probabilities of the respective values. Here $z_i$ is a binary variable that allows the background model $p(f_i|B)$ to explain those locations in the feature map $F$ that are in the foreground region $\mathcal{FG}$, but which the foreground model $(f_i|\mathfrak{N}_y, m)$ cannot explain well. A primary purpose of this mechanism is to make the cost function robust to partial occlusion. Figure 2 illustrates the inference process. Given an initial camera pose estimate we use the Neural Mesh Model to render a feature map $\bar{F}$ and evaluate the reconstruction loss in the foreground region $\mathcal{FG}$ (foreground score map), as well as the reconstruction error when using the background model only (background score map). Pixel-wised comparison of foreground score and background score yield the occlusion map $\mathcal{Z} = \{z_i \in \{0, 1\} | \forall i \in \mathcal{P}\}$. The map $Z$ indicates where feature vectors are explained by either the foreground or background model.

A fundamental benefit of our Neural Mesh Models is that, they are generative on the level of neural feature activations. This makes the overall reconstruction loss very smooth compared to related works who are generative on the pixel level. Therefore, NeMo can be optimized w.r.t. the pose parameters with standard stochastic gradient descent. We visualize the loss as a function of the individual pose parameters in Figure 2. Note that the losses are generally very smooth and contain one clear global optimum. This is in stark contrast to the optimization of classic generative models at the level of RGB pixels, which often requires complex hand designed initialization and optimization procedures to avoid the many local optima of the reconstruction loss Blanz & Vetter (2003); Schönborn et al. (2017).

## 4 Experiment

We first describe the experimental setup in Section 4.1. Subsequently, we study the performance of NeMo at 3D pose estimation in Section 4.2 and study the effect of crudely approximating the object geometry within NeMo single 3d cuboid, that one cuboid represent all object in each category, and multiple 3d cuboid, that one cuboid represent only one subtype of object in each category. We ablate the important modules of our model in Section 4.4.

### 4.1 Experimental Setup

**Evaluation.** The task of 3D object pose estimation involves the prediction of three rotation parameters (azimuth, elevation, in-plane rotation) of an object relative to the camera. In our eval-

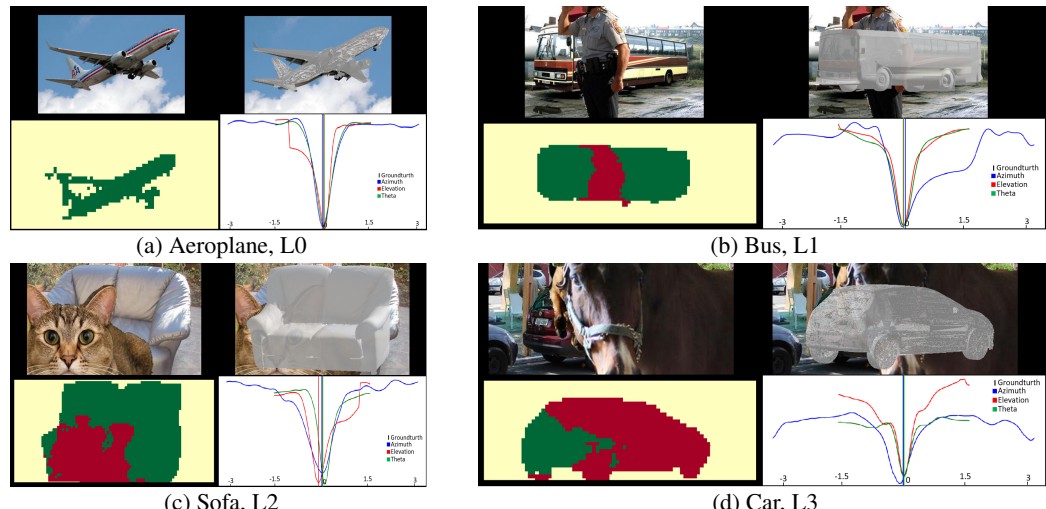

(a) Aeroplane, L0        (b) Bus, L1

(c) Sofa, L2        (d) Car, L3

Figure 3: Qualitative results of NeMo on PASCAL3D+ (L0) and occluded PASCAL3D+ (L1 & L2 & L3) for different categories under different occlusion level. For each example, we show four subfigures. Top-left: the input image; Top-right: A mesh superimposed on the input image in the predicted 3D pose. Bottom-left: The occluder localization result, where yellow is background, green is the non-occluded area of the object and red is the occluded area as predicted by NeMo. Bottom-right: The loss landscape for each individual camera parameter respectively. The colored vertical lines demonstrate the final prediction and the ground-truth parameter is at center of x-axis.

uation, we follow the protocol as proposed in related work Zhou et al. (2018) to measure the pose estimation error between the predicted rotation matrix and the ground truth rotation matrix: $\Delta\left(R_{pred}, R_{gt}\right) = \frac{\left\|\log m\left(R_{pred}^T R_{gt}\right)\right\|_F}{\sqrt{2}}$. We report two commonly used evaluation metrics the median of the rotation error, the percentage of predicted angles within a given accuracy threshold. Specifically, we use the thresholds $\frac{pi}{6}$ and $\frac{pi}{18}$. Following Zhou et al. (2018), we assume the centers and scales of the objects are given in all experiments.

**Datasets.** We evaluate NeMo on both the PASCAL3D+ dataset Xiang et al. (2014) and the occluded PASCAL3D+ dataset Wang et al. (2020). PASCAL3D+ contains 12 man-made object categories with 3D pose annotations and 3D meshes for each category respectively. We follow Wang et al. (2020) and Bai et al. (2020) to split the PASCAL3D+ into a training set with 11045 images and validation set with 10812 images. The occluded PASCAL3D+ dataset is a benchmark to evaluate robustness under occlusion. This dataset simulates realistic man-made occlusion by artificially superimposing occluders collected from the MS-COCO dataset Lin et al. (2014) on objects in PASCAL3D+. The dataset contains all 12 classes of objects from PASCAL3D+ dataset with three levels of occlusion, where L1: 20-40%, L2: 40-60%, L3: 60-80% of the object area are occluded.

We further test NeMo on the ObjectNet3D dataset Xiang et al. (2016), which is also a category-level 3D pose estimation benchmark. ObjectNet3D contains 100 different categories with 3D meshes, it contains totally 17101 training samples and 19604 testing samples, including 3556 occluded or truncated testing samples. Following Zhou et al. (2018), we report pose estimation results on 18 categories. Note that different from StarMap, we use all images during evaluation, including occluded or truncated samples.

**Training Setup.** In the training process, we use the 3D meshes (see Section 4.2 for experiments without the mesh geometry), the locations and scales of objects, and the 3D poses. We use Blender Community (2018) to reduce the resolution of the mesh because the meshes provided in PASCAL3D+ have a very high number of vertices. In order to balance the performance and computational cost, in particular the cost of the rendering process, we limit the size of the feature map produced by backbone to $\frac{1}{8}$ of the input image. To achieve this, we use the ResNet50 with two additional upsample layers as our backbone. We train a backbone for each category separately, and learn a Neural Mesh Model for each subtype in a category. We follow hyperparameter settings from Bai et al. (2020) for the contrastive loss. We train NeMo for 800 training epochs with a batch size of 108, which takes around 3 to 5 hours to train a category using 6 NVIDIA RTX Titan GPUs.

**Baselines.** We compare our model to StarMap Zhou et al. (2018) using their official implementation and training setup. Following common practice, we also evaluate a popular baseline that formu-

Table 1: Pose estimation results on PASCAL3D+ and the occluded PASCAL3D+ dataset. Occlusion level L0 are the orignal images from PASCAL3D+, while Occlusion Level L1 to L3 are the occluded PASCAL3D+ images with increasing occlusion ratio. We evaluate both baseline and NeMo using Accuracy (percentage, higher better) and Median Error (degree, lower better). Note that NeMo is exceptionally robust to partial occlusion.

| Evaluation Metric | $ACC_{\frac{\pi}{6}} \uparrow$ | | | | $ACC_{\frac{\pi}{18}} \uparrow$ | | | | $MedErr \downarrow$ | | | |
| --- | --- | --- | --- | --- | --- | --- | --- | --- | --- | --- | --- | --- |
| Occlusion Level | L0 | L1 | L2 | L3 | L0 | L1 | L2 | L3 | L0 | L1 | L2 | L3 |
| Res50-General | 88.1 | 70.4 | 52.8 | 37.8 | 44.6 | 25.3 | 14.5 | 6.7 | 11.7 | 17.9 | 30.4 | 46.4 |
| Res50-Specific | 87.6 | 73.2 | 58.4 | 43.1 | 43.9 | 28.1 | 18.6 | 9.9 | 11.8 | 17.3 | 26.1 | 44.0 |
| StarMap | **89.4** | 71.1 | 47.2 | 22.9 | 59.5 | 34.4 | 13.9 | 3.7 | 9.0 | 17.6 | 34.1 | 63.0 |
| NeMo | 84.1 | 73.1 | 59.9 | 41.3 | 60.4 | 45.1 | 30.2 | 14.5 | 9.3 | 15.6 | 24.1 | 41.8 |
| NeMo-MultiCuboid | 86.7 | **77.2** | **65.2** | **47.1** | **63.2** | **49.9** | **34.5** | **17.8** | **8.2** | **13.0** | **20.2** | **36.1** |
| NeMo-SingleCuboid | 86.1 | 76.0 | 63.9 | 46.8 | 61.0 | 46.3 | 32.0 | 17.1 | 8.8 | 13.6 | 20.9 | 36.5 |

lates pose estimation as a classification problem. In particular, we evaluate the performance of a deep neural network classifier that uses the same backbone as NeMo. We train a category specific Resnet50 (Res50-Specific), which formulates the pose estimation in each category as an individual classification problem. Furthermore, we train a non-specific Resnet50 (Res50-General), which performs pose estimation for all categories in a single classification task. We report the result of both architectures using the implementation provided by Zhou et al. (2018).

**Inference via Feature-level Rendering.** We implement the NeMo inference pipeline (see 3.4) using PyTorch3D Ravi et al. (2020). Specifically, we render the Neural Mesh Models into feature map $\bar{F}$ using the feature representations $\Theta$ stored at each mesh vertex. We estimate the object pose by minimizing the reconstruction loss as introduced in Equation 12. For initialization of pose optimization, we uniformly sample 144 different poses (12 azimuth angles, 4 elevation angles, 3 in-plane rotations respectively). Then we pick the initial pose with minimum reconstruction loss as a starting point of optimization (optimization conduct with only the chosen pose, others will be deprecated). On average each image takes about 8s with a single GPU for inference. The whole inference process on PASCAL3D+ takes about 3 hours using a 8 GPU machine.

## 4.2 ROBUST 3D POSE ESTIMATION UNDER OCCLUSION

**Baseline performances.** Table 1 (for categories specific scores, see 6) illustrates the 3D pose estimation results on PASCAL3D+ under different levels of occlusion. In the low accuracy setting ($ACC_{\frac{\pi}{6}}$) StarMap performs exceptionally well when the object is non-occluded ($L0$). However, with increasing level of partial occlusion, the performance of StarMap degrades massively, falling even below the basic classification models Res50-General and Res50-Specific. These results highlight that today's most common **deep networks for 3D pose estimation are not robust**. Similar, generalization patterns can be observed for the high accuracy setting ($ACC_{\frac{\pi}{18}}$). However, we can observe that the classification baselines do not perform as well as before, and hence are not well suited for fine-grained 3D pose estimation. Nevertheless, they outperform StarMap at high occlusion levels ($L2$ & $L3$).

**NeMo.** We evaluate NeMo in three different setups: NeMo uses a down-sampled object mesh as geometry representation, NeMo-MultiCuboid and NeMo-SingleCuboid approximate the 3D object geometry crudely using 3D cuboid boxes. We discuss the cuboid generation and results in detail in the next paragraph. Compared to the baseline performances, we observe that NeMo achieves competitive performance at estimating the 3D pose of non-occluded objects. Moreover, **NeMo is much more robust compared to all baseline approaches**. In particular, we observe that NeMo achieves the highest performance at every evaluation metric when the objects are partially occluded. Note that the training data for all models is exactly the same.

To further investigate and understand the robustness of NeMo, we qualitatively analyze the pose estimation and occluder location predictions of NeMo in Figure 3. Each subfigure shows the input image, the pose estimation result, the occluder localization map and the loss as a function of the pose angles. We visualize the loss landscape along each pose parameter (azimuth, elevation and in-plane rotation) by sampling the individual parameters in a fixed step size, while keeping all other parameters at their ground-truth value. We further split the binary occlusion map $\mathcal{Z}$ into three regions to highlight the occluder localization performance of NeMo. In particular, we split the region that is explained by the background model into a yellow and a red region. The red region is covered by rendered mesh and highlights the locations with the projected region of the mesh, which the neural

Table 2: Pose estimation results of ObjectNet3D. Evaluated via pose estimation accuracy percentage for error under $\frac{\pi}{6}$ (higher better). Both baseline and NeMo evaluated on all images of given category, including occluded and truncated. Overall, NeMo has higher accuracy in 14 categories while lower in 4 categories.

| $ACC_{\frac{\pi}{6}} \uparrow$ | bed | bookshelf | calculator | cellphone | computer | cabinet | guitar | iron | knife |
|---|---|---|---|---|---|---|---|---|---|
| StarMap | 40.0 | **72.9** | 21.1 | **41.9** | 62.1 | 79.9 | 38.7 | 2.0 | 6.1 |
| NeMo-MultiCuboid | **56.1** | 53.7 | **57.1** | 28.2 | **78.8** | **83.6** | **38.8** | **32.3** | **9.8** |
| $ACC_{\frac{\pi}{6}} \uparrow$ | microwave | pen | pot | rifle | slipper | stove | toilet | tub | wheelchair |
| StarMap | 86.9 | **12.4** | 45.1 | 3.0 | **13.3** | 79.7 | 35.6 | 46.4 | 17.7 |
| NeMo-MultiCuboid | **90.3** | 3.7 | **66.7** | **13.7** | 6.1 | **85.2** | **74.5** | **61.6** | **71.7** |

Table 3: Pose estimation results on PASCAL3D+ for objects in seen and unseen poses. The histogram on the left shows how we separate the PASCAL3D+ test dataset into subsets based on the azimuth pose of the object. We have similarly split the training dataset and trained all models only on the "seen" subset. We evaluate on both test sets (Seen & Unseen). Note the strong generalization performance of NeMo in unseen view points.

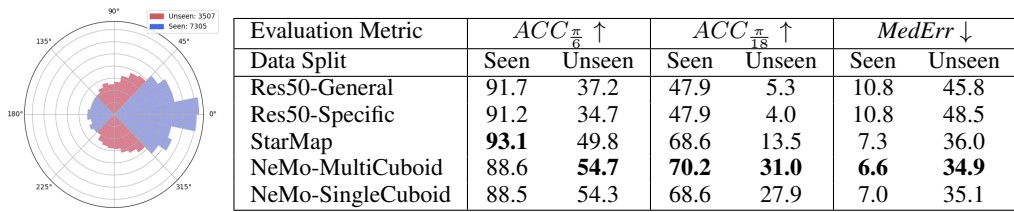

| Evaluation Metric | $ACC_{\frac{\pi}{6}} \uparrow$ | | $ACC_{\frac{\pi}{18}} \uparrow$ | | $MedErr \downarrow$ | |
|---|---|---|---|---|---|---|
| Data Split | Seen | Unseen | Seen | Unseen | Seen | Unseen |
| Res50-General | 91.7 | 37.2 | 47.9 | 5.3 | 10.8 | 45.8 |
| Res50-Specific | 91.2 | 34.7 | 47.9 | 4.0 | 10.8 | 48.5 |
| StarMap | **93.1** | 49.8 | 68.6 | 13.5 | 7.3 | 36.0 |
| NeMo-MultiCuboid | 88.6 | **54.7** | **70.2** | **31.0** | **6.6** | **34.9** |
| NeMo-SingleCuboid | 88.5 | 54.3 | 68.6 | 27.9 | 7.0 | 35.1 |

mesh model cannot explain well. Hence these mark the locations in the image that NeMo predicts to be occluded. From the qualitative illustrations, we observe that NeMo maintains high robustness even under extreme occlusion, when only a small part of the object is visible. Furthermore, we can clearly see that NeMo can approximately localize the occluders. This occluder localization property of NeMo makes our model not just more robust but also much more human-interpretable compared to standard deep network approaches.

**NeMo without detailed object mesh.** We approximate the object geometry in NeMo by replacing the downsampled mesh with 3D cuboid boxes (see Figure 5). The vertices of the cuboid meshes are evenly distributed on all six sides of the cuboid. For generating the cuboids, we use three constraint: 1) The cuboid should cover all the vertices of the original mesh with minimum volume; 2) The distances between each pair of adjacent vertices should be similar; 3) The total number of vertices for each mesh should be around 1200 vertices. We generate two different types of models. NeMo-MultiCuboid uses a separate cuboid for each object mesh in an object category, while NeMo-SingleCuboid uses on cuboid for all instances of a category.

We report the pose estimations results with NeMo using cuboid meshes in Table 1. The results show that approximating the detailed mesh representations of a category with **a single 3D cuboid gives surprisingly good results**. In particular, NeMo-SingleCuboid even often outperforms our standard model. This shows that generative models of neural network feature activations must not retain the detailed object geometry, because the feature activations are invariant to detailed shape properties. Moreover, NeMo-MultiCube outperforms the SingleCube model significantly. This suggests that for some categories the size between different sub-types can be very differnt (e.g. for the airplane class it could be a passenger jet or a fighter jet). Therefore, a single mesh may not be representative enough for some object categories. The MultiCuboid model even outperforms our the model with detailed mesh geometry. This is very likely caused by difficulties during the down-sampling of the original meshes in PASCAL3D+, which might remove important parts of the object geometry.

We also conduct experiment on ObjectNet3D dataset, which reported in Table 2. The result demonstrates that NeMo outperforms StarMap in 14 categories out of all 18 categories. Note that due to the considerable number of occluded and truncated images in ObjectNet3D dataset, this dataset is significantly harder than PASCAL3D+, however, NeMo still demonstrates reasonable accuracy.

## 4.3 GENERALIZATION TO UNSEEN VIEWS

To further investigate robustness of NeMo to out-of-distribution data, we evaluate the performance of NeMo when objects are observed from previously unseen viewpoints. For this, we split the PASCAL3D+ dataset into two sets based on the ground-truth azimuth angle. In particular, we use

Table 4: Ablation study on PASCAL3D+ and occluded PASCAL3D+. All ablation experiments are conducted with the NeMo-MultiCuboid model. The performance is reported in terms of Accuracy (percentage, higher better) and Median Error (degree, lower better).

| Evaluation Metric | $ACC_{\frac{\pi}{6}} \uparrow$ | | | | $ACC_{\frac{\pi}{18}} \uparrow$ | | | | $MedErr \downarrow$ | | | |
|---|---|---|---|---|---|---|---|---|---|---|---|---|
| Occlusion Level | L0 | L1 | L2 | L3 | L0 | L1 | L2 | L3 | L0 | L1 | L2 | L3 |
| NeMo | **86.7** | **77.3** | **65.2** | **47.1** | **63.2** | **49.2** | **34.5** | **17.8** | **8.2** | **13.1** | **20.2** | **36.1** |
| NeMo w/o outlier | 85.2 | 76.0 | 63.2 | 44.4 | 61.8 | 47.9 | 32.4 | 16.2 | 8.5 | 13.5 | 20.7 | 41.6 |
| NeMo w/o contrastive | 69.7 | 58.0 | 44.6 | 26.9 | 40.8 | 27.7 | 14.7 | 5.6 | 18.3 | 27.7 | 37.0 | 61.0 |

Table 5: Sensitivity of NeMo-MultiCuboid under different numbers of pose initializations during inference (Init Samples) on PASCAL3D+.

| Init Samples | $ACC_{\frac{\pi}{6}} \uparrow$ | $ACC_{\frac{\pi}{18}} \uparrow$ | $MedErr \downarrow$ |
|---|---|---|---|
| 144(Std.) | **86.7** | **63.2** | **8.2** |
| 72 | 86.3 | 63.0 | 8.3 |
| 36 | 84.1 | 61.1 | 8.8 |
| 12 | 81.2 | 57.7 | 9.3 |
| 6 | 80.4 | 57.7 | 9.6 |
| 1 | 54.9 | 38.9 | 35.6 |

the front and rear views for training. We evaluate all approaches on the full testing set and split the performance into seen (front and rear) and unseen (side) poses. The histogram on the left of Table 4 shows the distribution of ground-truth azimuth angles in the PASCAL3D+ test dataset. The seen-test-set contains 7305 images while the unseen-test-set contains 3507 images. Table 4 shows that NeMo can significantly better generalize to novel viewpoints compared to the baselines. For some categories the accuracy of NeMo on the unseen-test-set is even comparable to seen-test-set (Table 7). These results highlight the importance of building neural networks with 3D internal representations, which enable them to generalize exceptionally well to unseen 3D transformations.

## 4.4 ABLATION STUDY

In Table 4, we study the effect of each individual module of NeMo. Specifically, we remove the clutter feature, background score and occluder prediction during inference, and only use foreground score to calculate pose loss. This reduces the robustness to occlusion significantly. Furthermore, we remove the contrastive loss and use neural features that were extracted with an ImageNet-pretrained Resnet50 with non-parametric-upsampling. This leads to a massive decrease in performance, and hence highlights the importance of learning locally distinct feature representations. Table 5 (and Table 10) study the sensitivity of NeMo to the random pose initialization before the pose optimization. In this ablation, we evaluate NeMo-MultiCuboid with 144 down to 1 uniformly sampled initialization poses. Note that we do not run 144 optimization processes. We instead evaluate the reconstruction error for each initialization and start the optimization from the initializaiton with the lowest error. Hence, every experiment only involves one optimization run. The results demonstrate that NeMo benefits from the smooth lose landscape. With 6 initial samples NeMo achieves a reasonable performance, while 72 initial poses almost yield the maximum performance. This ablation clearly highlights that, unlike standard Render-and-Compare approaches Blanz & Vetter (1999); Schönborn et al. (2017), NeMo does not require complex designed initialization strategies.

## 5 CONCLUSION

In this work, we considered the problem of robust 3D pose estimation with neural networks. We found that **standard deep learning approaches do not give robust predictions** when objects are partially occluded or viewed from an unseen pose. In an effort to resolve this fundamental limitation we developed Neural Mesh Models (NeMo), a neural network architecture that **integrates a prototypical mesh representation with a generative model of neural features**. We combine NeMo with contrastive learning and show that this makes possible to **estimate the 3D pose with very high robustness to out-of-distribution data** using simple gradient-based render-and-compare. Our experiments demonstrate the superiority of NeMo compared to related work on a range of challenging datasets.

ACKNOWLEDGMENTS

We gratefully acknowledge funding support from ONR N00014-18-1-2119, ONR N00014-20-1-2206, the Institute for Assured Autonomy at JHU with Grant IAA 80052272, and the Swiss National Science Foundation with Grant P2BSP2.181713. We also thank Weichao Qiu, Qing Liu, Yutong Bai and Jiteng Mu for suggestions on our paper.

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

# A APPENDIX

Table 6: Pose estimation results on PASCAL3D+ (L0) for all categories respectively. Results reported in Accuracy (percentage, higher better) and Median Error (degree, lower better).

| | aero | bike | boat | bottle | bus | car | chair | table | mbike | sofa | train | tv | Mean |
|---|---|---|---|---|---|---|---|---|---|---|---|---|---|
| ↑ $ACC_{\frac{\pi}{6}}$ Res50-General | 83.0 | 79.6 | 73.1 | 87.9 | 96.8 | 95.5 | 91.1 | 82.0 | 80.7 | 97.0 | 94.9 | 83.3 | 88.1 |
| ↑ $ACC_{\frac{\pi}{6}}$ Res50-Specific | 79.5 | 75.8 | **73.5** | 90.3 | 93.5 | 95.6 | 89.1 | 82.4 | 79.7 | 96.3 | **96.0** | 84.6 | 87.6 |
| ↑ $ACC_{\frac{\pi}{6}}$ StarMap | **85.5** | **84.4** | 65.0 | **93.0** | **98.0** | 97.8 | **94.4** | **82.7** | **85.3** | **97.5** | 93.8 | **89.4** | **89.4** |
| ↑ $ACC_{\frac{\pi}{6}}$ NeMo | 73.3 | 66.4 | 65.5 | 83.0 | 87.4 | **98.8** | 82.8 | 81.9 | 74.6 | 94.7 | 87.0 | 85.5 | 84.1 |
| ↑ $ACC_{\frac{\pi}{6}}$ NeMo-MultiCuboid | 76.9 | 82.2 | 66.5 | 87.1 | 93.0 | 98.0 | 90.1 | 80.5 | 81.8 | 96.0 | 89.3 | 87.1 | 86.7 |
| ↑ $ACC_{\frac{\pi}{6}}$ NeMo-SingleCuboid | 82.2 | 78.4 | 68.1 | 88.0 | 91.7 | 98.2 | 87.0 | 76.9 | 85.0 | 95.0 | 83.0 | 82.2 | 86.1 |
| ↑ $ACC_{\frac{\pi}{18}}$ Res50-General | 31.3 | 25.7 | 23.9 | 35.9 | 67.2 | 63.5 | 37.0 | 40.2 | 18.9 | 62.5 | 51.2 | 24.9 | 44.6 |
| ↑ $ACC_{\frac{\pi}{18}}$ Res50-Specific | 29.1 | 22.9 | 25.3 | 39.0 | 62.7 | 62.9 | 37.5 | 42.0 | 19.5 | 57.5 | 50.2 | 25.4 | 43.9 |
| ↑ $ACC_{\frac{\pi}{18}}$ StarMap | **49.8** | 34.2 | 25.4 | **56.8** | **90.3** | 81.9 | **67.1** | 57.5 | 27.7 | **70.3** | 69.7 | 40.0 | 59.5 |
| ↑ $ACC_{\frac{\pi}{18}}$ NeMo | 39.0 | 31.3 | 29.6 | 38.6 | 83.1 | 94.8 | 46.9 | **58.1** | 29.3 | 61.1 | 71.1 | **66.4** | 60.4 |
| ↑ $ACC_{\frac{\pi}{18}}$ NeMo-MultiCuboid | 43.1 | **35.3** | 36.4 | 48.6 | 89.7 | **95.5** | 49.5 | 56.5 | **33.8** | 68.8 | **75.9** | 56.8 | **63.2** |
| ↑ $ACC_{\frac{\pi}{18}}$ NeMo-SingleCuboid | 49.7 | 29.5 | **37.7** | 49.3 | 89.3 | 94.7 | 49.5 | 52.9 | 29.0 | 58.5 | 70.1 | 42.4 | 61.0 |
| ↓ *MedErr* Res50-General | 13.3 | 15.9 | 15.6 | 12.1 | 8.9 | 8.8 | 11.5 | 11.4 | 16.6 | 8.7 | 9.9 | 15.8 | 11.7 |
| ↓ *MedErr* Res50-Specific | 14.2 | 17.3 | 15.4 | 11.7 | 9.0 | 8.8 | 12.0 | 11.0 | 17.1 | 9.2 | 10.0 | 14.9 | 11.8 |
| ↓ *MedErr* StarMap | **10.0** | 14.0 | 19.7 | **8.8** | 3.2 | 4.2 | **6.9** | 8.5 | 14.5 | **6.8** | 6.7 | 12.1 | 9.0 |
| ↓ *MedErr* NeMo | 13.8 | 17.5 | 18.3 | 12.8 | 3.4 | **2.7** | 10.7 | **8.2** | 16.1 | 8.0 | 5.6 | **6.6** | 9.3 |
| ↓ *MedErr* NeMo-MultiCuboid | 11.8 | **13.4** | **14.8** | 10.2 | **2.6** | 2.8 | 10.1 | 8.8 | **14.0** | 7.0 | **5.0** | 8.1 | **8.2** |
| ↓ *MedErr* NeMo-SingleCuboid | 10.1 | 16.3 | 14.9 | 10.2 | 3.2 | 3.2 | 10.1 | 9.3 | 14.1 | 8.6 | 5.4 | 12.2 | 8.8 |

Table 7: Pose estimation results on occluded PASCAL3D+ occlusion L1 for all categories respectively. Results reported in Accuracy (percentage, higher better) and Median Error (degree, lower better).

| | aero | bike | boat | bottle | bus | car | chair | table | mbike | sofa | train | tv | Mean |
|---|---|---|---|---|---|---|---|---|---|---|---|---|---|
| ↑ $ACC_{\frac{\pi}{6}}$ Res50-General | 57.3 | 56.8 | 51.4 | 78.3 | 82.5 | 80.0 | 62.3 | 63.1 | 61.1 | 84.9 | 87.8 | 69.8 | 70.4 |
| ↑ $ACC_{\frac{\pi}{6}}$ Res50-Specific | 54.0 | 59.5 | 48.9 | **84.4** | 86.1 | 84.4 | 67.1 | 64.9 | 65.9 | **87.8** | **92.4** | 74.5 | 73.2 |
| ↑ $ACC_{\frac{\pi}{6}}$ StarMap | 52.6 | 65.3 | 42.0 | 81.8 | **87.9** | 86.1 | 64.5 | 66.5 | 62.8 | 76.9 | 85.2 | 59.7 | 71.1 |
| ↑ $ACC_{\frac{\pi}{6}}$ NeMo | 49.0 | 51.4 | 52.9 | 73.5 | 82.2 | **94.3** | 70.2 | 67.9 | 53.8 | 86.7 | 75.0 | 79.4 | 73.1 |
| ↑ $ACC_{\frac{\pi}{6}}$ NeMo-MultiCuboid | 58.1 | **68.8** | **53.4** | 78.8 | 86.9 | 94.0 | 76.0 | **70.0** | 61.8 | 87.3 | 82.8 | **82.8** | **77.2** |
| ↑ $ACC_{\frac{\pi}{6}}$ NeMo-SingleCuboid | **61.9** | 63.4 | 52.9 | 81.3 | 84.8 | 92.7 | **78.4** | 68.2 | **68.9** | 87.1 | 80.3 | 76.9 | 76.0 |
| ↑ $ACC_{\frac{\pi}{18}}$ Res50-General | 11.8 | 12.5 | 12.3 | 26.5 | 45.0 | 40.7 | 14.7 | 22.3 | 10.7 | 24.4 | 34.9 | 13.0 | 25.3 |
| ↑ $ACC_{\frac{\pi}{18}}$ Res50-Specific | 12.4 | 10.7 | 13.8 | 30.2 | 46.9 | 44.8 | 21.2 | 24.0 | 10.4 | 28.0 | 40.6 | 17.9 | 28.1 |
| ↑ $ACC_{\frac{\pi}{18}}$ StarMap | 15.6 | 15.1 | 10.8 | 36.2 | 66.6 | 58.1 | 26.6 | 32.0 | 14.4 | 23.8 | 47.4 | 13.0 | 34.4 |
| ↑ $ACC_{\frac{\pi}{18}}$ NeMo | 18.5 | 19.9 | 19.1 | 24.0 | 72.1 | 82.0 | 25.8 | 35.7 | 12.6 | 44.3 | 54.0 | **49.0** | 45.1 |
| ↑ $ACC_{\frac{\pi}{18}}$ NeMo-MultiCuboid | 25.4 | **23.3** | 22.9 | 36.7 | **86.9** | **84.8** | 33.1 | **36.8** | 20.8 | **46.5** | **61.0** | 46.3 | **49.9** |
| ↑ $ACC_{\frac{\pi}{18}}$ NeMo-SingleCuboid | **29.3** | 18.0 | **24.3** | **41.5** | 76.1 | 80.5 | 27.2 | 31.4 | 19.4 | 39.9 | 55.1 | 32.0 | 46.3 |
| ↓ *MedErr* Res50-General | 25.3 | 24.5 | 29.0 | 14.9 | 10.6 | 11.2 | 22.4 | 18.1 | 23.3 | 15.5 | 11.7 | 21.1 | 17.9 |
| ↓ *MedErr* Res50-Specific | 26.8 | 23.7 | 31.0 | 13.8 | 10.5 | 10.6 | 18.2 | 16.7 | 21.8 | 13.6 | 10.9 | 19.3 | 17.3 |
| ↓ *MedErr* StarMap | 27.3 | 22.1 | 38.9 | 12.9 | 7.0 | 8.2 | 19.1 | 17.2 | 21.7 | 16.8 | 10.6 | 24.1 | 17.6 |
| ↓ *MedErr* NeMo | 30.8 | 29.0 | 27.3 | 17.6 | 5.9 | 5.1 | 18.6 | 14.7 | 27.4 | 11.3 | 8.8 | **10.2** | 15.6 |
| ↓ *MedErr* NeMo-MultiCuboid | 22.6 | **18.6** | **25.8** | 14.1 | **4.7** | **4.6** | **15.1** | **13.8** | 21.2 | **11.0** | **8.0** | 11.3 | **13.0** |
| ↓ *MedErr* NeMo-SingleCuboid | **18.9** | 23.2 | 26.7 | **12.6** | 5.2 | 5.4 | 15.6 | 15.4 | **20.1** | 12.1 | 8.6 | 15.3 | 13.6 |

Table 8: Pose estimation results on occluded PASCAL3D+ occlusion L2 for all categories respectively. Results reported in Accuracy (percentage, higher better) and Median Error (degree, lower better).

| | aero | bike | boat | bottle | bus | car | chair | table | mbike | sofa | train | tv | Mean |
|---|---|---|---|---|---|---|---|---|---|---|---|---|---|
| ↑ $ACC_{\frac{\pi}{6}}$ Res50-General | 33.3 | 40.2 | 33.6 | 70.6 | 69.5 | 57.0 | 41.8 | 47.4 | 43.3 | 66.8 | 80.4 | 58.1 | 52.8 |
| ↑ $ACC_{\frac{\pi}{6}}$ Res50-Specific | 36.3 | 44.9 | 36.1 | **76.1** | 73.1 | 65.5 | 53.2 | 49.5 | 45.4 | 72.7 | **88.3** | 65.0 | 58.4 |
| ↑ $ACC_{\frac{\pi}{6}}$ StarMap | 28.5 | 38.9 | 21.3 | 65.0 | 61.7 | 59.3 | 37.5 | 44.7 | 43.2 | 55.1 | 56.4 | 36.2 | 47.2 |
| ↑ $ACC_{\frac{\pi}{6}}$ NeMo | 38.2 | 41.2 | 39.6 | 58.3 | 72.6 | **84.7** | 50.7 | 51.1 | 34.9 | 70.1 | 60.0 | 64.6 | 59.9 |
| ↑ $ACC_{\frac{\pi}{6}}$ NeMo-MultiCuboid | 43.1 | **55.7** | 43.3 | 69.1 | **79.8** | 84.5 | 58.8 | **58.4** | 43.9 | 76.4 | 64.3 | **70.3** | **65.2** |
| ↑ $ACC_{\frac{\pi}{6}}$ NeMo-SingleCuboid | **43.4** | 49.6 | **43.6** | 76.0 | 71.2 | 83.8 | **61.9** | 55.9 | 50.9 | 78.3 | 63.1 | 68.6 | 63.9 |
| ↑ $ACC_{\frac{\pi}{18}}$ Res50-General | 6.1 | 4.5 | 7.2 | 20.1 | 25.9 | 21.4 | 9.5 | 13.2 | 6.1 | 14.0 | 23.0 | 8.6 | 14.5 |
| ↑ $ACC_{\frac{\pi}{18}}$ Res50-Specific | 5.7 | 6.9 | 8.0 | **25.5** | 33.9 | 29.1 | 13.0 | 11.6 | 6.8 | 18.4 | 32.0 | 13.8 | 18.6 |
| ↑ $ACC_{\frac{\pi}{18}}$ StarMap | 3.8 | 5.8 | 2.4 | 19.7 | 30.5 | 24.5 | 7.7 | 9.6 | 5.1 | 9.6 | 21.5 | 5.8 | 13.9 |
| ↑ $ACC_{\frac{\pi}{18}}$ NeMo | 10.7 | 10.5 | 11.3 | 13.9 | 55.8 | 60.6 | 9.3 | **20.3** | 6.3 | 26.1 | 34.6 | 32.1 | 30.2 |
| ↑ $ACC_{\frac{\pi}{18}}$ NeMo-MultiCuboid | 12.8 | **16.6** | **16.8** | 21.9 | **62.3** | **64.6** | 17.2 | 20.3 | **12.3** | **32.4** | **38.2** | **32.7** | **34.5** |
| ↑ $ACC_{\frac{\pi}{18}}$ NeMo-SingleCuboid | **14.9** | 11.1 | 15.6 | 18.2 | 56.0 | 62.4 | **17.4** | 18.7 | 10.2 | 30.5 | 36.4 | 22.4 | 32.0 |
| ↓ *MedErr* Res50-General | 49.3 | 42.5 | 58.5 | 17.7 | 15.9 | 21.3 | 35.4 | 32.0 | 36.1 | 20.3 | 15.2 | 25.3 | 30.4 |
| ↓ *MedErr* Res50-Specific | 45.8 | 33.9 | 52.8 | **16.3** | 12.4 | 15.1 | 27.1 | 30.9 | 32.4 | 18.3 | **12.3** | 24.1 | 26.1 |
| ↓ *MedErr* StarMap | 55.2 | 37.1 | 69.1 | 20.6 | 19.0 | 21.3 | 39.2 | 34.0 | 35.5 | 27.0 | 24.8 | 40.3 | 34.1 |
| ↓ *MedErr* NeMo | 39.8 | 37.7 | 44.2 | 24.8 | 8.8 | 7.7 | 29.7 | 28.5 | 47.5 | 16.9 | 18.2 | 17.0 | 24.1 |
| ↓ *MedErr* NeMo-MultiCuboid | **38.5** | **26.4** | **38.2** | 18.8 | **7.0** | 7.3 | 23.0 | **23.0** | 36.0 | **14.0** | 14.9 | **16.1** | **20.2** |
| ↓ *MedErr* NeMo-SingleCuboid | 39.9 | 30.6 | 38.8 | 19.5 | 8.3 | 7.8 | **21.3** | 24.8 | **29.5** | 14.2 | 16.9 | 18.5 | 20.9 |

Table 9: Pose estimation results on occluded PASCAL3D+ occlusion L3 for all categories respectively. Results reported in Accuracy (percentage, higher better) and Median Error (degree, lower better).

| | aero | bike | boat | bottle | bus | car | chair | table | mbike | sofa | train | tv | Mean |
|---|---|---|---|---|---|---|---|---|---|---|---|---|---|
| ↑ $ACC_{\frac{\pi}{6}}$ Res50-General | 18.3 | 20.8 | 21.2 | 62.1 | 57.0 | 36.9 | 31.1 | 32.2 | 24.3 | 56.2 | 64.5 | 53.4 | 37.8 |
| ↑ $ACC_{\frac{\pi}{6}}$ Res50-Specific | 20.0 | 33.4 | 25.5 | **67.5** | **57.8** | 42.0 | 40.7 | 33.9 | 30.3 | 56.6 | **82.8** | **56.5** | 43.1 |
| ↑ $ACC_{\frac{\pi}{6}}$ StarMap | 7.6 | 18.5 | 10.6 | 46.3 | 35.1 | 25.3 | 22.5 | 24.6 | 15.9 | 26.4 | 24.0 | 19.5 | 22.9 |
| ↑ $ACC_{\frac{\pi}{6}}$ NeMo | **24.0** | 31.3 | 27.4 | 43.3 | 48.8 | 62.8 | 31.8 | 29.7 | 18.4 | 44.2 | 34.5 | 51.4 | 41.3 |
| ↑ $ACC_{\frac{\pi}{6}}$ NeMo-MultiCuboid | 23.8 | **34.3** | 29.5 | 53.9 | 56.0 | **65.5** | 43.4 | **41.5** | 25.4 | 58.2 | 43.2 | 54.1 | **47.1** |
| ↑ $ACC_{\frac{\pi}{6}}$ NeMo-SingleCuboid | 20.6 | 33.8 | 27.6 | 61.7 | 49.9 | 61.8 | **44.7** | 41.2 | 35.3 | 62.9 | 47.9 | 50.2 | 46.8 |
| ↑ $ACC_{\frac{\pi}{18}}$ Res50-General | 1.6 | 2.3 | 2.9 | 11.9 | 14.4 | 7.6 | 3.8 | 5.7 | 3.1 | 7.9 | 12.7 | 8.9 | 6.7 |
| ↑ $ACC_{\frac{\pi}{18}}$ Res50-Specific | 2.0 | 5.5 | 4.8 | **16.7** | 21.1 | 13.1 | 5.9 | 5.7 | 4.3 | 9.9 | **22.5** | 6.0 | 9.9 |
| ↑ $ACC_{\frac{\pi}{18}}$ StarMap | 0.8 | 1.7 | 1.1 | 11.8 | 8.3 | 4.8 | 2.1 | 2.6 | 1.6 | 2.8 | 5.2 | 0.7 | 3.7 |
| ↑ $ACC_{\frac{\pi}{18}}$ NeMo | 4.4 | 6.2 | 6.7 | 6.8 | 26.5 | 31.1 | 3.4 | 6.7 | 2.0 | 9.3 | 13.0 | **16.7** | 14.5 |
| ↑ $ACC_{\frac{\pi}{18}}$ NeMo-MultiCuboid | **5.5** | 5.2 | 7.9 | 10.8 | **34.2** | **37.4** | 7.4 | 8.2 | 4.5 | 15.8 | 15.1 | 15.9 | **17.8** |
| ↑ $ACC_{\frac{\pi}{18}}$ NeMo-SingleCuboid | 4.7 | **6.7** | **8.6** | 11.7 | 29.2 | 33.7 | **11.0** | 10.7 | 4.9 | **17.8** | 17.2 | 10.9 | 17.1 |
| ↓ *MedErr* Res50-General | 69.8 | 70.9 | 73.2 | 22.7 | 24.9 | 46.7 | 41.5 | 44.4 | 59.8 | 26.3 | 21.3 | 28.4 | 46.4 |
| ↓ *MedErr* Res50-Specific | 65.8 | 47.1 | 75.8 | **20.9** | **18.5** | 46.6 | 35.9 | 49.9 | 56.3 | 26.4 | **15.3** | 26.5 | 44.0 |
| ↓ *MedErr* StarMap | 87.0 | 67.6 | 90.2 | 32.6 | 51.3 | 64.0 | 60.7 | 53.2 | 73.4 | 51.0 | 52.7 | 54.7 | 63.0 |
| ↓ *MedErr* NeMo | **65.3** | 48.4 | 65.2 | 34.5 | 34.9 | 17.2 | 44.6 | 55.7 | 74.3 | 33.7 | 47.6 | 29.3 | 41.8 |
| ↓ *MedErr* NeMo-MultiCuboid | 69.8 | 49.6 | **63.0** | 28.2 | 19.4 | **14.9** | 35.4 | 39.9 | 60.0 | 23.7 | 38.1 | 27.2 | **36.1** |
| ↓ *MedErr* NeMo-SingleCuboid | 74.8 | **46.1** | 70.1 | 24.5 | 30.2 | 16.3 | **35.2** | **37.5** | 50.5 | **21.5** | 31.7 | 29.9 | 36.5 |

Table 10: Full table for 5. This table shows category specific results of NeMo-MultiCuboid pose estimation performance on PASCAL3D+ using different number of initialization pose during inference. The Init Samples shows total number of initialization pose e.g. 144 means we uniformly sample 12(azimuth) * 4(elevation) * 3(in-plane rotation) poses. Std. mean this setting is standard settings and used in main experiment.

| Category | | aero | bike | boat | bottle | bus | car | chair | table | mbike | sofa | train | tv | Mean |
|---|---|---|---|---|---|---|---|---|---|---|---|---|---|---|
| $\uparrow ACC_{\frac{\pi}{6}}$ | 144(Std.) | 76.9 | **82.2** | **66.5** | **87.1** | **93.0** | **98.0** | **90.1** | 80.5 | 81.8 | **96.0** | **89.3** | 87.1 | **86.7** |
| | 72 | **77.1** | 81.9 | 64.6 | 86.5 | 93.0 | 98.0 | 89.2 | 81.3 | **82.2** | 95.8 | 85.9 | 87.0 | 86.3 |
| | 36 | 74.6 | 79.2 | 60.0 | 86.6 | 89.8 | 94.7 | 88.6 | 79.5 | 80.0 | 95.2 | 86.4 | 86.5 | 84.1 |
| | 12 | 73.4 | 78.1 | 57.1 | 86.2 | 79.9 | 86.9 | 90.1 | **81.6** | 79.3 | 94.6 | 82.5 | 86.5 | 81.2 |
| | 6 | 69.7 | 78.1 | 58.3 | 85.7 | 82.5 | 90.9 | 87.8 | 68.6 | 80.3 | 95.0 | 79.4 | 86.0 | 80.4 |
| | 1 | 38.7 | 33.6 | 34.2 | 86.9 | 54.9 | 40.6 | 77.5 | 68.3 | 27.8 | 89.7 | 78.6 | 84.7 | 54.9 |
| $\uparrow ACC_{\frac{\pi}{18}}$ | 144(Std.) | 43.1 | 35.3 | 36.4 | **48.6** | 89.7 | **95.5** | 49.5 | 56.5 | 33.8 | 68.8 | **75.9** | **56.8** | 63.2 |
| | 72 | **43.2** | **35.7** | **36.6** | 47.5 | **89.8** | 95.2 | 48.7 | 56.7 | **34.0** | **69.1** | 72.6 | 56.6 | 63.0 |
| | 36 | 41.3 | 33.2 | 31.6 | 47.8 | 85.5 | 91.8 | 49.5 | 56.1 | 33.0 | 68.4 | 74.1 | 56.3 | 61.1 |
| | 12 | 41.1 | 32.2 | 26.6 | 47.3 | 73.3 | 84.4 | **49.7** | **57.0** | 33.5 | 67.9 | 67.6 | 55.8 | 57.7 |
| | 6 | 38.3 | 32.1 | 30.5 | 46.9 | 78.4 | 88.2 | 48.1 | 46.7 | 33.0 | 68.1 | 66.9 | 55.5 | 57.7 |
| | 1 | 22.7 | 19.7 | 21.6 | 47.4 | 44.7 | 37.9 | 44.6 | 47.3 | 14.6 | 65.5 | 62.1 | 54.5 | 38.9 |
| $\downarrow MedErr$ | 144(Std.) | **11.8** | **13.4** | **14.8** | **10.2** | **2.6** | **2.8** | 10.1 | 8.8 | **14.0** | **7.0** | **5.0** | **8.1** | **8.2** |
| | 72 | 11.9 | 13.4 | 15.7 | 10.4 | 2.6 | 2.8 | 10.3 | **8.7** | 14.0 | 7.0 | 5.2 | 8.2 | 8.3 |
| | 36 | 12.4 | 14.5 | 19.1 | 10.4 | 2.8 | 2.9 | 10.1 | 8.8 | 14.2 | 7.0 | 5.0 | 8.2 | 8.8 |
| | 12 | 12.5 | 14.8 | 22.6 | 10.5 | 3.8 | 3.1 | **10.0** | 8.7 | 14.5 | 7.1 | 5.7 | 8.3 | 9.3 |
| | 6 | 14.2 | 14.8 | 21.4 | 10.5 | 3.1 | 2.9 | 10.4 | 11.5 | 14.3 | 7.1 | 5.6 | 8.4 | 9.6 |
| | 1 | 49.0 | 77.0 | 63.9 | 10.4 | 27.7 | 43.6 | 11.1 | 11.3 | 78.0 | 7.4 | 6.0 | 8.7 | 35.6 |

Table 11: Experiment for NeMo-MultiCuboid when subtype is not given during inference. In the w/o subtype experiment we use NMMs of all subtypes to do inference on each image respectively, then pick the predicted pose with subtypes with minimum reconstruction loss. The result demonstrate that distinguishing subtypes is not necessary for pose estimation with NeMo.

| | $\uparrow ACC_{\frac{\pi}{6}}$ | aero | bike | boat | bottle | bus | car | chair | table | mbike | sofa | train | tv | Mean |
|---|---|---|---|---|---|---|---|---|---|---|---|---|---|---|
| L0 | with subtype | 76.9 | **82.2** | 66.5 | **87.1** | 93.0 | 98.0 | **90.1** | **80.5** | 81.8 | **96.0** | 89.3 | **87.1** | **86.7** |
| | w/o subtype | **77.7** | 78.3 | **70.9** | 82.6 | **94.5** | **98.7** | 86.4 | 75.5 | **83.7** | 93.8 | 89.3 | 81.0 | 85.8 |
| L1 | with subtype | 58.1 | **68.8** | 53.4 | **78.8** | 86.9 | 94.0 | **76.0** | **70.0** | 61.8 | **87.3** | 82.8 | **82.8** | **77.2** |
| | w/o subtype | **59.4** | 63.4 | **56.0** | 74.9 | **90.3** | **96.1** | 73.4 | 63.0 | **64.7** | 83.8 | **84.0** | 77.6 | 76.5 |
| L2 | with subtype | 43.1 | **55.7** | 43.3 | **69.1** | 79.8 | 84.5 | 58.8 | **58.4** | 43.9 | **76.4** | 64.3 | **70.3** | **65.2** |
| | w/o subtype | **46.3** | 48.1 | **44.3** | 67.3 | **84.0** | **86.2** | **61.5** | 50.9 | **45.1** | 70.2 | **67.0** | 67.4 | 64.7 |
| L3 | with subtype | 23.8 | **34.3** | 29.5 | 53.9 | 56.0 | 65.5 | 43.4 | **41.5** | 25.4 | 58.2 | 43.2 | **54.1** | 47.1 |
| | w/o subtype | **26.1** | 28.6 | **31.5** | **54.9** | **61.2** | **66.9** | **44.3** | 34.4 | 25.0 | 54.0 | **49.2** | 53.8 | **47.2** |

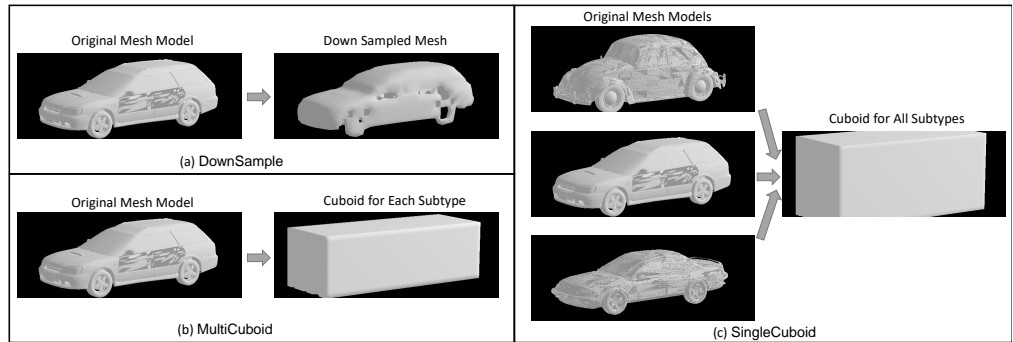

Figure 5: Using detailed mesh model we can create all type of mesh models for NeMo. (a) We use remesh method in Blender to down sample the original mesh. The processed mesh contains 1722 vertices. (b) Following rules in 4.2, we create subtype specificed cuboid (one cuboid for each subtype), which used in NeMo-MultiCuboid approach. The cuboid contains 1096 vertices. (c) We create the subtype general cuboid by requiring the cuboid cover original meshes of all subtypes. And we use the created cuboid to represent all objects in this category, which reported as NeMo-SingleCuboid. This cuboid contains 1080 vertices.

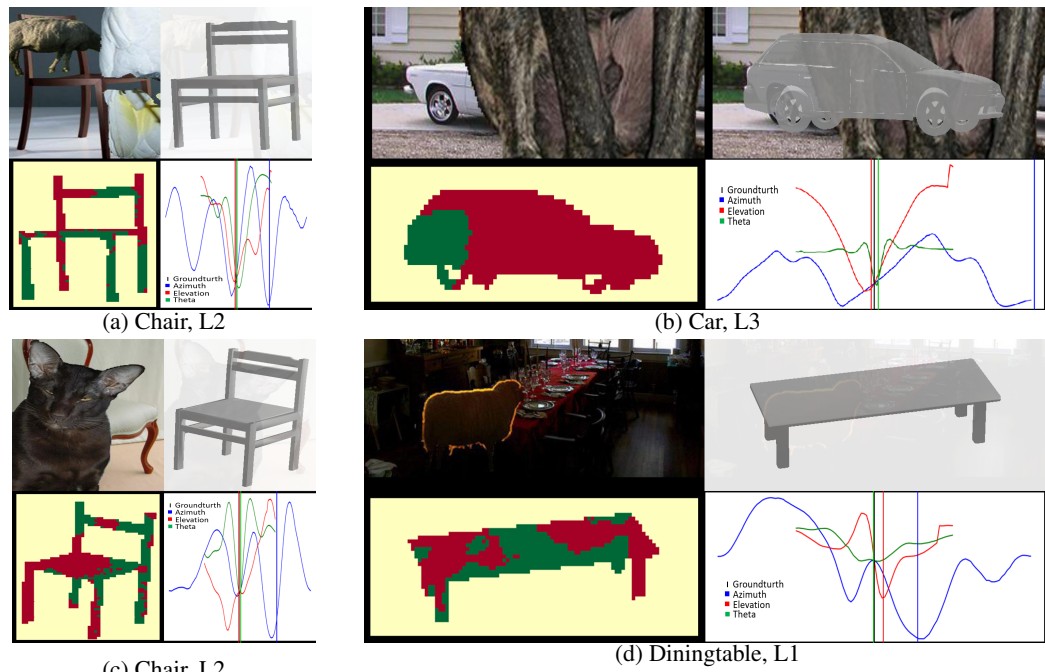

Figure 6: Visualization of failure case of NeMo on occluded PASCAL3D+. For each example, we show four subfigures. Top-left: the input image; Top-right: A mesh superimposed on the input image in the predicted 3D pose. Bottom-left: The occluder localization result, where yellow is background, green is the non-occluded area of the object and red is the occluded area as predicted by NeMo. Bottom-right: The loss landscape for each individual camera parameter respectively. The colored vertical lines demonstrate the final prediction and the ground-truth parameter is at center of x-axis.

Table 12: Pose estimation results on PASCAL3D+ under unseen pose for CAR category. Figure shows the distribution of azimuth in PASCAL3D+ testing set of car category and our splitting.

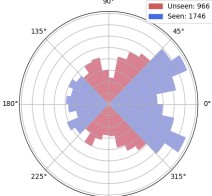

| Evaluation Metric | $ACC_{\frac{\pi}{6}}$ ↑ | | $ACC_{\frac{\pi}{18}}$ ↑ | | ↓ *MedErr* ↓ | |
|---|---|---|---|---|---|---|
| Data Split | Seen | Unseen | Seen | Unseen | Seen | Unseen |
| Res50-General | 97.2 | 55.3 | 72.2 | 11.5 | 8.1 | 25.5 |
| Res50-Specific | 97.2 | 52.5 | 70.5 | 11.7 | 8.2 | 27.7 |
| StarMap | **98.2** | 77.6 | 93.7 | 34.2 | 3.4 | 15.5 |
| NeMo-MultiCuboid | 96.8 | 97.0 | 94.8 | **85.4** | **2.6** | **5.2** |
| NeMo-SingleCuboid | 98.0 | **97.8** | **96.3** | 78.8 | 2.9 | 5.9 |

