# OpenReview forum: "NeMo: Neural Mesh Models of Contrastive Features for Robust 3D Pose Estimation"
_ICLR.cc/2021/Conference — ICLR 2021 Poster_

### Official Review · AnonReviewer4 · 2020-10-27
**Neat combination of differentiable rendering and contrastive feature learning, but experiments can be improved.**

**Rating:** 6
**Confidence:** 4

**Review:**

Summary:
The paper presents a novel approach for 3d pose estimation by combining render-and-compare (analysis-by-synthesis) and contrastive feature learning. The key idea is to render and compare learned latent features instead of synthesized RGB colors to optimize 6D pose parameters. The proposed method learns latent feature vectors on a template mesh as well as target images via backbone neural networks such that matched regions have similar features while latent features are as distinctive as possible. The paper evaluates the novel formulation on  PASCAL3D+, the occluded PASCAL3D+, and ObjectNet3D dataset, demonstrating the render-and compare optimization with the proposed approach is more robust to appearance change and partial occlusions.

Although the paper presents very interesting idea, the paper leaves several critical aspects unclear (please see the comments below). Thus, I believe the paper is not ready for publication.

Pros:

\- The paper addresses the sensitivity of render-and-compare approaches by leveraging the learned latent features that are agnostic to appearance change.

\- Partial occlusion is handled by jointly predicted binary mask, allowing the render-and-compare optimization focuses on visible regions.

\- The evaluation shows that contrastive feature learning for distinctiveness plays a critical role to robust optimization.


Cons:

\- As the authors mentioned in the related work, the traditional R&C approaches are sensitive to initialization. However, this paper misses any form of ablation study on the sensitivity on initialization, which is critical for practical use.

\- One major limitation of the proposed method is to rely on a fixed template model. As the size and shape of objects in the same category often vary, it is not clear why not to incorporate simple shape parametrization in the template model (via PCA or axis aligned scale factors) as in traditional R&C methods.

\- The experiments in Sec. 4.2. do not fully support the advantage of NeMo over the baselines especially in case of partial occlusions. What if we provide explicit segmentation mask for each baseline as in NeMo? If the performance becomes comparable, to make deep networks more robust, we would need only explicit segmentation, not necessarily render-and-Compare optimization with contrastive feature learning.


Questions/Remarks:

\- How sensitive is the proposed approach to the choice of differentiable renderer?

\- As the entire pipeline can be trained end-to-end, the proposed method could incorporate pose alignment process as part of objective functions during feature learning process. Why is such an objective function not incorporated? Also I’m wondering if such an optimization-driven feature learning would eliminate the need of contrastive feature learning.

\- The paper explains how to compute z_i only in the caption of Fig. 2. I would recommend explaining it in the main text.

\- As the latent features on the mesh is fixed throughout the optimization once it’s trained, calling it a generative model would not be appropriate.

\- In Sec. 3.2., it’s not clear how binary mixture coefficients \alpha_k is used.

================================================================
Post Rebuttal:

Thanks for the revision and detailed rebuttal. I think the clarity is largely improved now. As the paper has sufficient technical novelty with fairly clear description, I'm slightly inclined towards acceptance. However, I would recommend further clarifying the following point.

>We want to point out that NeMo has exactly the same amount of supervision as all other methods
- The fact that NeMo jointly reason occlusions in an unsupervised manner was not clearly explained. As the authors described, this property makes difference in terms of robustness to partial occlusions. Clarification on this in the text is very helpful.

---

> ### Author Response · Authors · 2020-11-17
> **Re: Questions from Reviewer 4**
>
> **As the authors mentioned in the related work, the traditional R&C approaches are sensitive to initialization. However, this paper misses any form of ablation study on the sensitivity on initialization, which is critical for practical use.**
> We decribe the initialization strategy in the Common questions part, and we also conduct an additional ablation study about this.
>
> **One major limitation of the proposed method is to rely on a fixed template model. As the size and shape of objects in the same category often vary, it is not clear why not to incorporate simple shape parametrization in the template model (via PCA or axis aligned scale factors) as in traditional R&C methods.**
> We thank the reviewer for pointing out this highly relevant point that we should have addressed in more detail. Traditional R&C methods, such as 3D Morphable Models, are learned by bringing high resolution 3D meshes into dense correspondence, after which the main modes of variation can be learned using PCA. The registration process needed to achieve dense correspondence is highly complex. An important (and possibly surprising) insight of our experimental results is that such complex PCA models are not required for estimating the 3D pose of an object accurately. Rather a simple, fixed, box-like template is sufficient. The key advantage of NeMo over traditional R&C methods is that NeMo does R&C on a feature representation that is learned to be invariant to shape deformations. Whereas approaches such as 3DMMs have to model the detailed shape deformations because they do R&C on the pixel intensities. Note that pixel intensities are highly variant to even small shape deformations. We expect that combining NeMo with shape deformation models will be required for vision tasks that are more fine-grained compared to 3D pose estimation, e.g. such as detailed part segmentation. We think this would be a very interesting future research direction for extending NeMo.
>
> **The experiments in Sec. 4.2. do not fully support the advantage of NeMo over the baselines especially in case of partial occlusions. What if we provide explicit segmentation mask for each baseline as in NeMo? If the performance becomes comparable, to make deep networks more robust, we would need only explicit segmentation, not necessarily render-and-Compare optimization with contrastive feature learning.**
> We want to point out that NeMo has exactly the same amount of supervision as all other methods. We do not provide a human supervised segmentation mask. The generative nature of NeMo enables it to find out by itself which parts of the object are visible and which parts are occluded. This is a core advantage of NeMo over the baseline approaches, which enables it to naturally become robust to partial occlusion without the need for additional supervision.
>
> **How sensitive is the proposed approach to the choice of differentiable renderer?**
> We think using a different type of differentiable renderer to implement NeMo should have a similar result, though the choice of renderer might affect the inference efficiency. We use PyTorch3D to implement NeMo because PyTorch3D is easy to integrate with PyTorch, which allows us easily integrate the feature extractor and pose optimizer.
>
> **As the entire pipeline can be trained end-to-end, the proposed method could incorporate pose alignment process as part of objective functions during feature learning process. Why is such an objective function not incorporated? Also I’m wondering if such an optimization-driven feature learning would eliminate the need of contrastive feature learning.**
> This is a very interesting question. What we aim to achieve with the contrastive feature learning, is to make the features distinct from each other. This serves as a proxy task for making the overall reconstruction loss smooth. Alternatively, as the reviewer suggests, we could directly try to optimize the reconstruction loss to be convex and to have a global optimum at the correct position in the parameter space. However, this would require to run the optimization process while training the feature extractor. It is unclear to us how this could be computationally feasible.

---

> ### Author Response · Authors · 2020-11-17
> **Continue Re: Questions from Reviewer 4**
>
> **The paper explains how to compute z_i only in the caption of Fig. 2. I would recommend explaining it in the main text.**
> Thanks for the suggestion. We have updated this in the revision.
>
> **As the latent features on the mesh is fixed throughout the optimization once it’s trained, calling it a generative model would not be appropriate.**
> This is also a very interesting question. In fact, the latent features on the NeMo mesh define the mean vectors of a Gaussian distribution with fixed variance (Equation 3). We can sample from these distributions to obtain new features for every vertex in the model and subsequently render those sampled features into a feature map. Hence NeMo is a generative model of feature maps. During inference, sampling from the distribution is not required because for the maximum likelihood estimation of the pose parameters, it is sufficient to minimize the distance to the mean representation.
>
> **In Sec. 3.2., it’s not clear how binary mixture coefficients \alpha_k is used.**
> Thanks for pointing out the incorrect \alpha_k, which is not used in our final version of the model. We removed it in the revision.

---

### Official Review · AnonReviewer1 · 2020-10-29
**Novel method to estimate object pose which works well also with occlusion**

**Rating:** 7
**Confidence:** 4

**Review:**

The authors propose a novel 3D neural mesh model of objects that is generative. They demonstrate that standard deep learning approaches to 3D pose estimation are highly sensitive to partial occlusion. Since their method works in a render and compare manner, it enables the method to be more robust to artifacts in general and partial occlusion in particular. They also achieve a highly competitive 3D pose estimation performance on popular dataset. They go on to show that even very crude prototypical approximation of the object geometry using a cuboid.

The neural mesh model representation is novel and clearly leads to superior robustness. Also the contrastive loss is clever and effective. Finally even when the 3D models are approximated by just cuboids, the results are very competitive. Their experiments are exhaustive and convincing. I would vote for acceptance of the paper.

However, the writing has a lot of scope for improved. E.g. it is difficult to understand how exactly the 3D mesh is converted to its neural mesh representation, etc. For understanding how much is a 3D model approximated when using its corresponding low resolution cubiod, a figure which shows a few classes, their 3D models and also their low resolution decimated 3D model will be helpful.

---

> ### Author Response · Authors · 2020-11-17
> **Re: Questions from Reviewer 1**
>
> **It is difficult to understand how exactly the 3D mesh is converted to its neural mesh representation, etc. For understanding how much is a 3D model approximated when using its corresponding low resolution cubiod, a figure which shows a few classes, their 3D models and also their low resolution decimated 3D model will be helpful.**
> We thank the reviewer for point out the ambiguity in our description. We make it more clear in the revision. We have a figure (Figure 5 in the appendix) that shows how the cuboid and low resolution mesh look. But due to the space limitations, we haven’t put this figure in the main text.

---

### Official Review · AnonReviewer2 · 2020-11-01
**Interesting direction but concerns regarding empirical setup**

**Rating:** 7
**Confidence:** 4

**Review:**

This paper tackles the task of pose prediction and takes a render-and-compare approach. However, instead of rendering pixel colors, the key insight is to render features -- each mesh vertex is associated with 3D (learned) features which are encouraged to match computed 2D image features. This 'neural mesh' representation allows pose inference via SGD as one can optimize for pose s.t. the rendered features best match the image features, and is also robust to foreground occlusion. The paper demonstrates results on ObjectNet3d and PASCAL3D+ where the proposed approach is shown to be more robust to occlusion and also better at precise pose estimation.



**Strengths**
I feel the overall approach proposed here is novel, interesting and elegant. While prior render-and-compare based pose estimation approaches have been proposed, they typically rely on semantic keypoints. This work essentially 'densifies' the notion of keypoints to any mesh vertex and instead of a one-hot identity vector, associates a semantic feature with it. The visualizations and the error landscapes shown do convince the reader that this allows accurate inference and gives robustness to occlusion for free.

The empirical results and ablations (assuming concerns below are addressed) evaluate the method across two datasets and occlusion settings, and show clear improvements over existing SOTA. The approach is also well ablated e.g. the necessity of the contrastive term is shown, and the benefits of handling outliers as means for robustness to occlusion are also analyzed.

It is also encouraging to see that the method works even when using a coarse geometry e.g. a cuboid as the reference mesh and this in particular shows the benefits of having learned features.


**Concerns**
I have several concerns/questions regarding the empirical setup and results, and feel that these have not been detailed sufficiently in the text. On this aspect, I feel the paper lacks clarity and would hope to see these answered before recommending acceptance.

- Each category in PASCAL3D (or Objectnet3D) has different 'subtypes'. The text in 'training setup' states 'we .. learn a NMM for each subtype separately'. This raises several questions:
a) Are separate Neural Mesh Models for subtypes learned even in the 'single-cuboid' case i.e. is only the geometry the same and features learned separately for each subtype?
b) More importantly, how is the subtype known at inference? The optimization objective in Eqn 12 assumes a known neural mesh model, but if there are many possible subtypes, it is not explained how this is inferred given a new image.

- Related to the above point, the optimization in Eq 12 also assumes a known 'FB/BG' mask. It is also not described how this is known for a new image. Is an additional predictor used?

- The text does not describe how the optimization for inference is performed? Does simple gradient descent with a fixed starting point work? Or are many different/random starting points used? Also, how efficient is inference?

- The failure cases are not analyzed/highlighted. For example, all the qualitative examples are shown for cases where the template matches the object rather well. What happens if this is not true e.g. for chairs?

- One result I am very puzzled by is the 'NeMo w/o contrastive' case. In this case, why don't all the learned features  (both 2D and 3D) tend to a constant? While this is shown to be worse than the full method, I am surprised this works at all! For example, Eqn 12 would be optimal if all f_i and N_y are zeros and all poses would be equally optimal! I would be curious on why this does not happen - is there some other objective term / inductive bias that prevents this.

- As a relatively minor point, while the empirical results show gain over 'StarMap' in cases with severe occlusion, this method is similar in case of the entire dataset (L0) without additional cropped occluders. If the information available to all methods at inference is the same, then this is not a major concern.

---------

Overall, while I feel the paper presents an interesting idea, there are just too many unknowns w.r.t. the inference setup (and also some surprising results) for me to recommend acceptance. If these can be clarified in the response and the comparisons are indeed fair to the baselines, I would be happy to update my rating.

--------------------------------------
Updates after author response:
---------------------------------------
I think the revisions and the responses did address all the concerns I had, in particular towards assuring that the method and baselines leverage the same information for inference. Additionally, the experiments where one can 'search' for the optimal subtype instead of assuming known subtype at inference also showed encouraging results.

Overall, I think the paper writing and presentation is much improved, and I would argue for acceptance as the paper presents a simple and intuitive idea which is shown to work (rather surprisingly!) well.

---

> ### Author Response · Authors · 2020-11-17
> **Re: Questions from Reviewer 2**
>
> **Each category in PASCAL3D (or Objectnet3D) has different 'subtypes'. The text in 'training setup' states 'we .. learn a NMM for each subtype separately'. This raises several questions: a) Are separate Neural Mesh Models for subtypes learned even in the 'single-cuboid' case i.e. is only the geometry the same and features learned separately for each subtype? b) More importantly, how is the subtype known at inference? The optimization objective in Eqn 12 assumes a known neural mesh model, but if there are many possible subtypes, it is not explained how this is inferred given a new image.**
> We thank the reviewer for pointing out the ambiguity in our writing when we discuss the settings of subtypes.
> a) For the ‘SingleCuboid’ setting, we do not distinguish between subtypes (we assume there is only one subtype for each category). And thus, we only learn one NMM for each category (features are the same for each subtype).
> b) For the ‘DownSample’ and ‘MultiCuboid’ settings, we assume the subtype is known. When the subtype is not given, one feasible approach is to run an optimization with all subtypes and then pick the predicted pose with minimum reconstruction loss. Note that this process will induce a longer inference time as the pose needs to be estimated with all NMMs. On the other hand, we observe that distinguishing subtypes is not necessary for pose estimation with NeMo. We conduct an experiment using such an inference strategy on occlusion L0, L1 and L2:
> | pi/6                            | aeroplane | bicycle | boat | bottle | bus  | car  | chair | diningtable | motorbike | sofa | train | tvmonitor |   mean |
> |---------------------------------|-----------|---------|------|--------|------|------|-------|-------------|-----------|------|-------|-----------|------|
> | L0 NeMo-MultiCuboid             | 76.9      | 82.2    | 66.5 | 87.1   | 93.0 | 98.0 | 90.1  | 80.5        | 81.8      | 96.0 | 89.3  | 87.1      | 86.7 |
> | L0 NeMo-MultiCuboid w/o subtype | 77.7      | 78.3    | 70.9 | 82.6   | 94.5 | 98.7 | 86.4  | 75.5        | 83.7      | 93.8 | 89.3  | 81.0 | 85.8 |
> | L1 NeMo-MultiCuboid             | 58.1      | 68.8    | 53.4 | 78.8   | 86.9 | 94.0 | 76.0  | 70.0        | 61.8      | 87.3 | 82.8  | 82.8      | 77.2 |
> | L1 NeMo-MultiCuboid w/o subtype | 59.4      | 63.4    | 56.0 | 74.9   | 90.3 | 96.1 | 73.4  | 63.0        | 64.7      | 83.8 | 84.0  | 77.6 | 76.5 |
> | L2 NeMo-MultiCuboid             | 43.1      | 55.7    | 43.3 | 69.1   | 79.8 | 84.5 | 58.8  | 58.4        | 43.9      | 76.4 | 64.3  | 70.3      | 65.2 |
> | L2 NeMo-MultiCuboid w/o subtype | 46.3      | 48.1    | 44.3 | 67.3   | 84.0 | 86.2 | 61.5  | 50.9        | 45.1      | 70.2 | 67.0  | 67.4      |  64.7 |
>
> **Related to the above point, the optimization in Eq 12 also assumes a known 'FB/BG' mask. It is also not described how this is known for a new image. Is an additional predictor used?**
> Thanks for pointing out the unclearness of Eq 12. We do not require an additional predictor. The FG mask is simply the area covered by the projected NMM (non-zero area in \bar{F}). The BG mask is the remaining area (zero area in \bar{F}).
>
> **The text does not describe how the optimization for inference is performed? Does simple gradient descent with a fixed starting point work? Or are many different/random starting points used? Also, how efficient is inference?**
> We detailly described the initialization strategy in the Common questions part, and we also conduct some ablation studies about this. For inference time, on average, each image takes about 8s with a single GPU for inference. The whole inference process on PASCAL3D+ (10812 images) will take about 3 hours using one 8 GPU machine.
>
> **The failure cases are not analyzed/highlighted. For example, all the qualitative examples are shown for cases where the template matches the object rather well. What happens if this is not true e.g. for chairs?**
> In our revision, we put the visualization of the failure cases in Appendix Figure 6.

---

> ### Author Response · Authors · 2020-11-17
> **Continue Re: Questions from Reviewer 2**
>
> **One result I am very puzzled by is the 'NeMo w/o contrastive' case. In this case, why don't all the learned features (both 2D and 3D) tend to a constant? While this is shown to be worse than the full method, I am surprised this works at all! For example, Eqn 12 would be optimal if all f_i and N_y are zeros and all poses would be equally optimal! I would be curious on why this does not happen - is there some other objective term / inductive bias that prevents this.**
> In the ablation study 'NeMo w/o contrastive', we use the Imagenet pre-trained backbone as the feature extractor. During training, we fix the backbone and use the same update strategy as proposed in the paper. Note that the features from ImageNet pre-trained networks have a minor distinguishing ability for parts (as this work shows [1]). This means the learned features are not an average of random vectors and will not be a constant. (No other objective term / inductive bias here.) However, this minor distinguishability is limited and easily affected by pose changes. And the induced contrastive loss will help these features be invariant to pose changes, which allows us to do the pose estimation more accurately.
> [1] J. Wang, C. Xie, Z. Zhang, J. Zhu, L. Xie, and A. Yuille. Detecting semantic parts on partially occluded objects. In British Machine Vision Conference, 2017
>
> **As a relatively minor point, while the empirical results show gain over 'StarMap' in cases with severe occlusion, this method is similar in case of the entire dataset (L0) without additional cropped occluders. If the information available to all methods at inference is the same, then this is not a major concern.**
> We argue that because StarMap uses a keypoint based pose estimation approach, which is not robust when part of keypoints are occluded. To explain this, we also evaluated the keypoint detection accuracy of StarMap under different levels of occlusion, as the following table shows.
> | Occlusion Level | L0     | L1     | L2     | L3     |
> |-----------------|--------|--------|--------|--------|
> | Mean PCK=0.1    | 0.8477 | 0.6006 | 0.3634 | 0.1796 |
>
> From the above Table we can see that the accuracy of the keypoint prediction of StarMap is significantly affected by occlusion. Note that for keypoint based approaches, the pose estimation is highly depend on the quality of keypoints predictions. On the other hand, NeMo uses a dense representation in a render-and-compare manner, with a specially designed strategy to discard features that are distorted by occlusion. We argue that these two characteristics allow NeMo to retain a significantly higher robustness.
> Yes, both method are training on unoccluded images, and use object position and scale during inference.

---

### Official Review · AnonReviewer3 · 2020-11-02
**Reasonable paper: Marginally above acceptance threshold.**

**Rating:** 6
**Confidence:** 3

**Review:**

This paper describes a neural mesh renderer that operates on a feature level for 3D object pose estimation. The optimization surface of the loss between the render (for a set of pose parameters) with the actual object image is improved using comparison in the feature space compared to RGB pixel space. The 3D CAD model of the object is converted to a mesh, which is converted to feature space - one feature per 3D mesh vertex. Contrastive learning of foreground vs background mesh features further improves the optimization landscape, even in the presence of occlusion. Experiments performed on the occluded Pascal3D+ dataset and the ObjectNet3D datasets illustrate the utility of conversion to feature space, and outperform some baselines. The paper is reasonably well written, but could do with a spell-check. Contrastive loss and learning in feature space is not novel, but their application to a mesh model for render and compare-type object pose estimation seems to be new. It could do with comparisons with some other state of the art render and compare baselines like [Chen et. al. 20] Category Level Object Pose Estimation via Neural Analysis-by-Synthesis.

---

> ### Author Response · Authors · 2020-11-17
> **Re: Questions from Reviewer 3**
>
> **It could do with comparisons with some other state of the art render and compare baselines like [Chen et. al. 20] Category Level Object Pose Estimation via Neural Analysis-by-Synthesis.**
> We thank the reviewer for the suggestion on potential baselines. Unfortunately, the Code of the suggested paper seems not to been published. However, we would also not expect that the renderer would have a significant impact on our performance.

---

### Author Response · Authors · 2020-11-17
**Reply to all reviewers**

First, we thank all reviewers for their efforts in reading and reviewing our submission. We also thank the reviewers for the detailed comments, questions, and suggestions.

## Update
1) We fix linguistic errors and ambiguities in the model section.
2) We add an extensive experiment about pose estimation under unseen pose. The experiment results demonstrate that NeMo can significantly better generalize to previously unseen poses compared to the baselines. Please refer to Section 4.3 in our revised version for more details.
3) An additional ablation study regarding the sensitivity of NeMo to the initialization before the pose optimization during inference.

## Common Question
**How sensitive is NeMo to initialization? Does simple gradient descent with a fixed starting point work? How many different/random starting points are used?**
In the reported experiments, we uniformly sample 144 starting poses (12 azimuth samples from 0 to 360°, 4 elevation samples for -30° to 60°, 3 theta samples from -30° to 30°). We choose the initial pose with minimum reconstruction loss and then start the optimization process. Hence we only run one optimization for every image, but we test several different random poses before running the optimization. As reviewer 2 and reviewer 4 suggested, we add an ablation study about the sensitivity of initialization.
We also show the results in this review in the Table below. The results demonstrate that unlike standard Render-and-Compare approaches, NeMo does not require a complex designed initialization strategy. Six initial samples allow NeMo to work reasonably well. Increasing the number of initial samples does further improve performance.
| Init Sample      | pi/6 | pi/18 | Med|
| 144(Standard) | 86.7 | 63.2  | 8.2  |
| 72                      | 86.3 | 63.0  | 8.3  |
| 36                      | 84.1 | 61.1  | 8.8  |
| 12                      | 81.2 | 57.7  | 9.3  |
| 6                        | 80.4 | 57.7  | 9.6  |
| 1                        | 54.9 | 38.9  | 35.6 |

---

### Decision · Program_Chairs · 2021-01-07
**Final Decision**

**Decision:**

Accept (Poster)

**Comment:**

This paper received 4 reviews with mixed initial ratings: 4,5,6,7. The main concerns of R2 and R4, who gave unfavorable scores, included lack of clarity around design choices and inconclusiveness of some of the experiments. In response to that, the authors submitted a new revision with a summary of changes and provided detailed responses to each of the reviews, which seemed to have addressed these concerns. R2 and R4 upgraded their scores and recommended acceptance.
As a result, the final recommendation is to accept for presentation at ICLR as a poster.